# 19.31% binary organic solar cell and low non-radiative recombination enabled by non-monotonic intermediate state transition

Jiehao Fu [1,2], Patrick W. K. Fong [1,2], Heng Liu[3], Chieh-Szu Huang [4], Xinhui Lu [3], Shirong Lu[5], Maged Abdelsamie[6,7], Tim Kodalle [8], Carolin M. Sutter-Fella [8], Yang Yang [4] ✉ & Gang Li [1,2] ✉

Non-fullerene acceptors based organic solar cells represent the frontier of the field, owing to both the materials and morphology manipulation innovations. Non-radiative recombination loss suppression and performance boosting are in the center of organic solar cell research. Here, we developed a non-monotonic intermediate state manipulation strategy for state-of-the-art organic solar cells by employing 1,3,5-trichlorobenzene as crystallization regulator, which optimizes the film crystallization process, regulates the self-organization of bulk-heterojunction in a non-monotonic manner, i.e., first enhancing and then relaxing the molecular aggregation. As a result, the excessive aggregation of non-fullerene acceptors is avoided and we have achieved efficient organic solar cells with reduced non-radiative recombination loss. In PM6:BTP-eC9 organic solar cell, our strategy successfully offers a record binary organic solar cell efficiency of 19.31% (18.93% certified) with very low non-radiative recombination loss of 0.190 eV. And lower non-radiative recombination loss of 0.168 eV is further achieved in PM1:BTP-eC9 organic solar cell (19.10% efficiency), giving great promise to future organic solar cell research.

Tremendous efforts in non-fullerene acceptor (NFA) materials have brought organic solar cells (OSCs) to a new era[1-3]. Now, the reported power conversion efficiency (PCE) of OSCs has exceeded 18%[4-11], with 18.2% certified record on NREL efficiency chart (https://www.nrel.gov/pv/cell-efficiency.html). However, compared to perovskite counterparts at 25.7% PCE, there is still clear PCE gap mainly due to the high non-radiative recombination loss in OSCs[12]. Therefore, a key question is how to form high-quality organic active layer that can reduce non-radiative recombination loss without deteriorating charge separation

and transport. In the field of OSC, the quality of active layer is related to the distribution and molecular stacking of donor and acceptor (D:A) bulk-heterojunction (BHJ) blend[13-15], which is more complicated than that in perovskite. In our previous fullerene-based work, we demonstrated that slowing down evaporation rate of high boiling point solvent is a strategy to induce highly ordered and crystalline polymer in the blend film[15,16]. Some of the landmark fullerene acceptor OSC works like solvent annealing, additive strategies are also tightly linked to BHJ active layer drying and crystallization kinetics[17,18], but due to the

[1]Department of Electronic and Information Engineering, Research Institute for Smart Energy (RISE), Photonic Research Institute (PRI), Guangdong-Hong Kong-Macao Joint Laboratory for Photonic-Thermal-Electrical Energy Materials and Devices, The Hong Kong Polytechnic University, Hung Hom, Kowloon, Hong Kong 999077, China. [2]The Hong Kong Polytechnic University Shenzhen Research Institute, Shenzhen 518057, China. [3]Department of Physics, The Chinese University of Hong Kong, Shatin, Hong Kong 999077, China. [4]Department of Materials Science and Engineering, University of California Los Angeles (UCLA), Los Angeles, CA 90095, USA. [5]School of Materials Science and Engineering, Taizhou University, Taizhou 318000, China. [6]Materials Sciences Division, Lawrence Berkeley National Laboratory, Berkeley, CA 94720, USA. [7]Materials Science and Engineering Department, King Fahd University of Petroleum and Minerals, Dhahran, Saudi Arabia. [8]Molecular Foundry, Lawrence Berkeley National Laboratory, Berkeley, CA 94720, USA. ✉e-mail: yangy@ucla.edu; gang.w.li@polyu.edu.hk

limitation of characterization tools, little insightful detail about the drying and crystallization kinetics of active layer has been presented so far. Entering NFA OSC era, the morphology-modifying techniques are commonly restricted to molecule optimization and ternary strategy, assisted by solvent additives originally designed for fullerene-based OSC systems, which does not fully realize the potential of NFA[4,5,19–24]. Taking the benchmark additive DIO as an example, DIO can increase the crystallinity of NFA while it has less impact on polymer donor[25–27]. The low volatility of DIO leads to the excessive aggregation of NFA associated with increased non-radiative recombination loss of OSC, which is one reason why DIO treated devices commonly suffer more serious voltage loss than as-cast devices[8,28]. While the recent explorations on volatile solid additives have led to better device stability comparing to the traditional solvent additives[26,27,29], little obviously enhanced PCE over solvent additives approach has been validated. Therefore, it is imperative to develop new morphology-regulating techniques that can optimize the self-organization of D:A and reduce non-radiative recombination simultaneously. Here, by employing 1,3,5-trichlorobenzene (TCB) as crystallization regulator[30–32], we report a non-monotonic intermediate state manipulation (ISM) strategy to maneuver the self-organization process of D:A blend. TCB can interact with both polymer donor and NFA, thereby improving their crystallinity and contributing to more efficient and balanced charge transport process. At the same time, the volatility of TCB is excellent, and it can be removed during spin-coating process. Finally, the OSC morphology of D:A molecules experienced a two-step manipulation process—the first enhancement and then relaxation of molecular aggregation. Assisted by the delicate intermediate states transition, active layer with more suitable molecular aggregation is achieved, facilitating higher $EQE_{EL}$, eventually leading to a record PCE of 19.31% (18.93% certified) in binary OSC with reduced $E_{loss,nr}$ of 0.190 eV. And lower $E_{loss,nr}$ of 0.168 eV is further achieved in high efficiency PM1:BTP-eC9 OSC (with

19.10% PCE). With the combined high device efficiency, reduced non-radiative recombination, excellent generality, and superior stability, the ISM strategy provides a new promising route towards OSC technology future.

## Results

### Interaction between TCB and active materials

The chemical structures of PM6, TCB, and Y6 are presented in Fig. 1a–c. We first conducted differential scanning calorimetry (DSC) test, formerly used to identify eutectic phase in OSC[33]. As illustrated in Fig. 1d–f, on the one hand, for PM6, Y6, and PM6:Y6 samples, no peak is observed in the cooling direction in a temperature range of 90-10°C, illustrating no endothermic or exothermic behavior in these samples. On the other hand, neat TCB shows an obvious exothermic peak at 53°C, which is related to the solidification temperature of TCB during the cooling process. The most interesting thermal behavior occurs in samples of TCB:PM6, TCB:Y6, and TCB:PM6:Y6. Take TCB:PM6 as an example, except the exothermic peak (at around 53°C) that attributed to the solidification process of TCB, we can observe another exothermic peak at lower temperature. As the neat PM6 sample does not show any peak during the cooling process, we ascribe this unexpected peak to the formation of a new phase in TCB:PM6 complex—to be specific, TCB can interact with PM6 and form a new phase. The similar phenomenon is again observed in TCB:Y6 and TCB:PM6:Y6 complexes, revealing that TCB can simultaneously interact with both donor and acceptor materials. To further elucidate the underlying mechanism at molecular level, we performed density functional theory (DFT) simulation (Supplementary Fig. 1). According to the computational result, in TCB molecule, the hydrogen (H) atoms show the maximum positive electrostatic potential (ESP) value. While oxygen (O) atoms in carbonyl groups (-CO) and nitrogen (N) atoms in cyano groups (-CN) show the minimum negative ESP value in PM6 molecule and Y6 molecule,

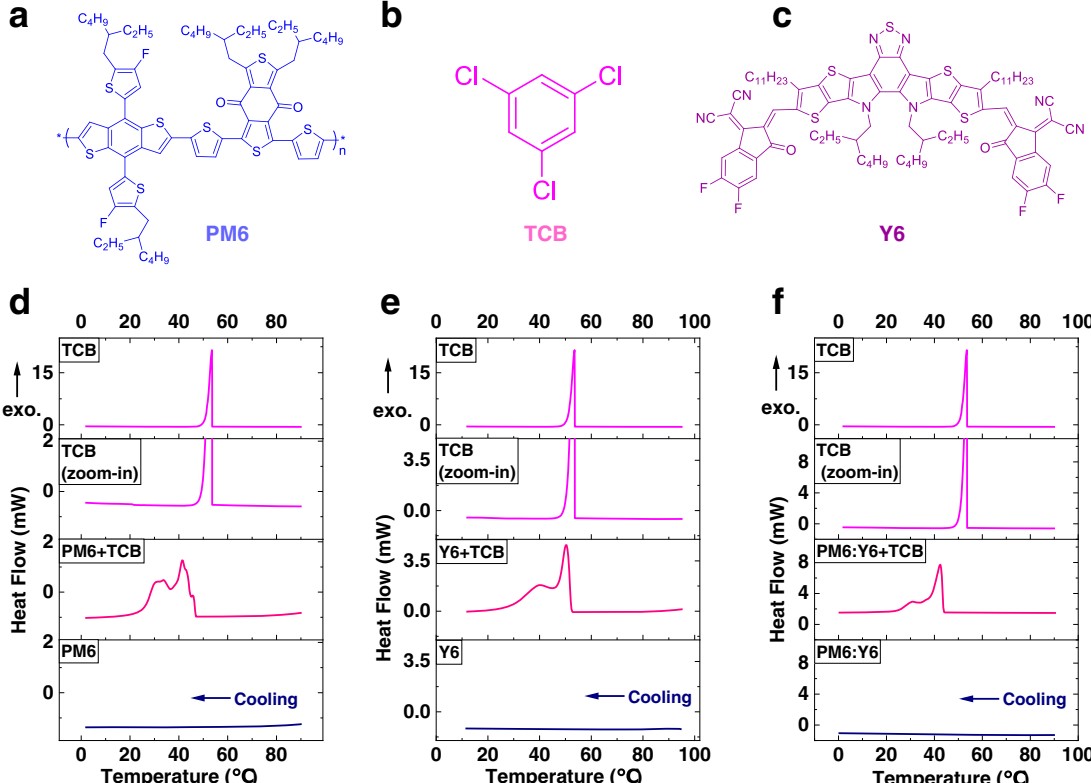

**Fig. 1 | Chemical structures and thermal behaviors between TCB and active materials.** Chemical structures of PM6 (**a**), TCB (**b**), and Y6 (**c**). **d** DSC thermograms (cooling process) of PM6, PM6:TCB, and TCB. **e** DSC thermograms (cooling process) of Y6, Y6:TCB, and TCB. **f** DSC thermograms (cooling process) of PM6:Y6, PM6:Y6:TCB, and TCB. Here exo. is the abbreviation of exothermic.

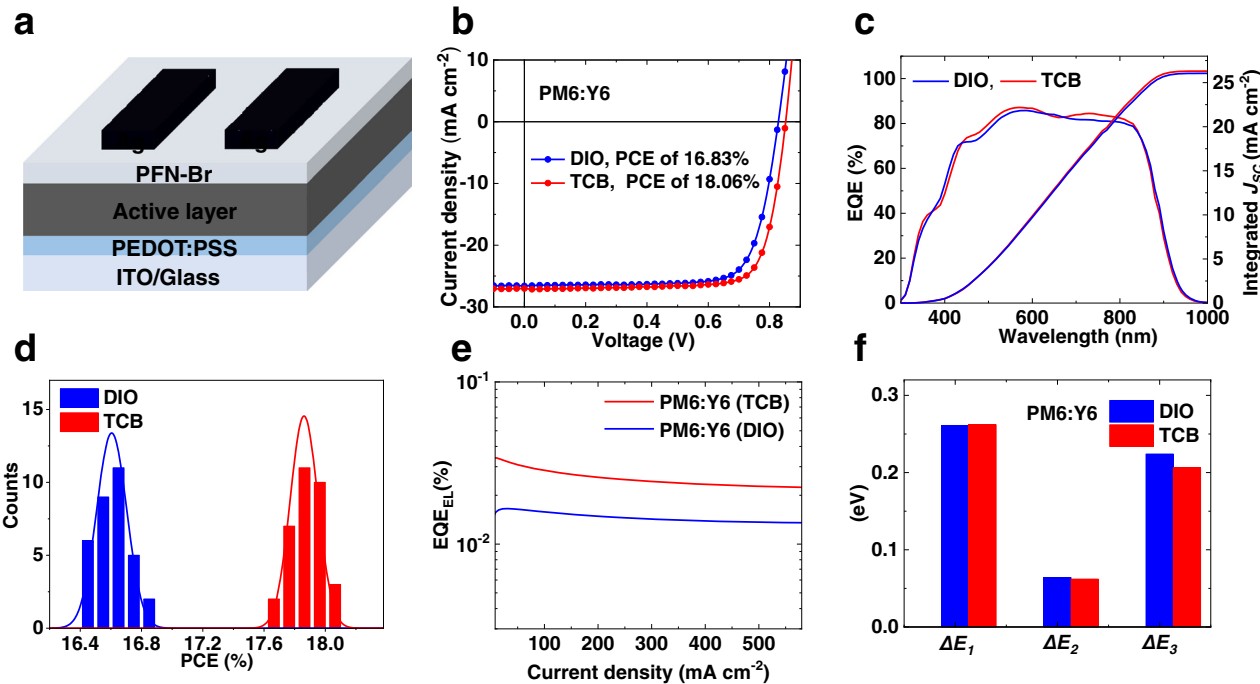

**Fig. 2 | Device performance of OSCs with DIO and TCB processing. a** Device structure used in this work. **b** $J–V$ curves for PM6: Y6-based OSCs with benchmark solvent additive DIO and with TCB. **c** EQE spectra for PM6:Y6-based OSCs with DIO and with TCB. **d** PCE histograms of PM6:Y6-based OSCs with DIO and with TCB. **e** $EQE_{EL}$ of PM6:Y6 devices with different treatments at various injected current densities. **f** Detailed energy loss in the DIO processed and TCB processed PM6:Y6 devices. Source data are provided as a Source Data file.

**Table 1 | Detailed photovoltaic performances of PM6:Y6-based devices processed with different treatments**

| Additive | $V_{OC}$ (V) | $J_{SC}$ (mA cm$^{-2}$) | $J_{SC}^{cal}$ (mA cm$^{-2}$) | FF (%) | PCE[a] (%) |
| --- | --- | --- | --- | --- | --- |
| DIO | 0.829 | 26.60 | 26.06 | 76.36 | 16.83 (16.61 ± 0.10) |
| TCB | 0.852 | 27.02 | 26.31 | 78.43 | 18.06 (17.86 ± 0.09) |

[a]The average PCEs with standard deviation were calculated from 33 devices in each case.
[cal]Integrated $J_{SC}$ values from EQE measurements.

respectively. This indicates the interaction between TCB and light absorbing materials originates from hydrogen bonds, i.e. -CO...H- and -CN...H-. In our recent work, we found eutectic phase behavior between the additive and acceptor (happened at the heating process) can be used as the driving force for nanomorphology optimization of absorbing layer[33]. In this work, no extra peak can be observed, except the melting peak of TCB in heating process (Supplementary Fig. 2). However very interestingly, in the cooling process, all three (PM6, Y6, D:A blend) mixtures with TCB exhibited new phases, which has never been reported in OSCs before. In principle, the crystallization-related physical processes are potentially effective in influencing OSC active layer film formation/crystallization kinetics, thereby influencing device performance. Therefore, we examined whether the interaction between TCB and active materials could have a positive impact on device performance.

### The impact of TCB on device performance

We fabricated OSCs with sandwich structure of indium tin oxide (ITO) / poly (3,4-ethylenedioxythiophene):poly (styrenesulfonate) (PEDOT:PSS)/PM6:Y6/poly[(9,9-bis(3'-((N,N-dimethyl)-N-ethyl ammonium)-propyl)−2,7-fluorene)-alt-2,7-(9,9-dioctylfluorene)] dibromide (PFN-Br) / Ag (Fig. 2a). The current density–voltage ($J–V$) curves of optimal OSCs treated with TCB and the current OSC field benchmark solvent additive 1,8-diiodooctane (DIO) are plotted in Fig. 2b, and the corresponding photovoltaic parameters are summarized in Table 1. Compared to the device without additive treatment with 16.16% PCE (Supplementary

Table 1), the DIO treated devices show a champion PCE of 16.83%, with a $V_{OC}$ of 0.829 V, a $J_{SC}$ of 26.60 mA/cm$^2$ and a FF of 76.39%. Excitingly, for the TCB-treated devices, the $V_{OC}$ of champion device increases to 0.852 V, and the other two parameters also get improvements—$J_{SC}$ and FF increase to 27.02 mA/cm$^2$ and 78.43%, respectively, thereby leading to a PCE of 18.06%. It is worth mentioning that 18.06% is among the highest efficiency for PM6:Y6-based binary system so far. To verify the $J_{SC}$ from $J–V$ test, we conducted external quantum efficiency (EQE) measurements (Fig. 2c). The integrated $J_{SC}$ values are 26.06 and 26.31 mA/cm$^2$ for the DIO and TCB-treated devices, respectively, which are in good agreement with the results form $J–V$ test.

### Physical properties of devices with different treatments

To understand the much higher efficiency improvement in the TCB-treated device from a physical point of view, we then analyzed the charge transport and recombination processes. The charge carrier transport properties were investigated by space charge limited current (SCLC) method. As shown in Supplementary Fig. 3 and Supplementary Table 3, TCB treatment contributes to similar electron ($3.5 \times 10^{-4}$ cm$^2$/Vs in DIO device versus $3.6 \times 10^{-4}$ cm$^2$/Vs in the TCB processed device) and slightly faster hole ($2.5 \times 10^{-4}$ cm$^2$/Vs in DIO device versus $3.0 \times 10^{-4}$ cm$^2$/Vs in the TCB processed device) mobility, which should ascribe to the more ordered molecular stacking in the TCB processed film (will be discussed later). Transient photovoltage (TPV) measurement was performed to explore the charge recombination dynamics. As shown in Supplementary Fig. 4, device with TCB

**Table 2 | Detailed $E_{loss}$ parameters of PM6:Y6 systems made with different treatments**

| Active layer | $V_{OC}$[a] (V) | $E_{loss}$ (eV) | $E_g$ (eV) | $\Delta E_1$ (eV) | $\Delta E_2$ (eV) | $EQE_{EL}$ (%) | $\Delta E_3$ (eV) | $\Delta E_3^{cal}$ (eV) |
|---|---|---|---|---|---|---|---|---|
| PM6:Y6-DIO | 0.833 | 0.558 | 1.391 | 0.261 | 0.064 | $1.7 \times 10^{-2}$ | 0.224 | 0.233 |
| PM6:Y6-TCB | 0.858 | 0.538 | 1.396 | 0.262 | 0.062 | $3.4 \times 10^{-2}$ | 0.206 | 0.214 |

[a]Device area is ~11 mm², no mask applied.

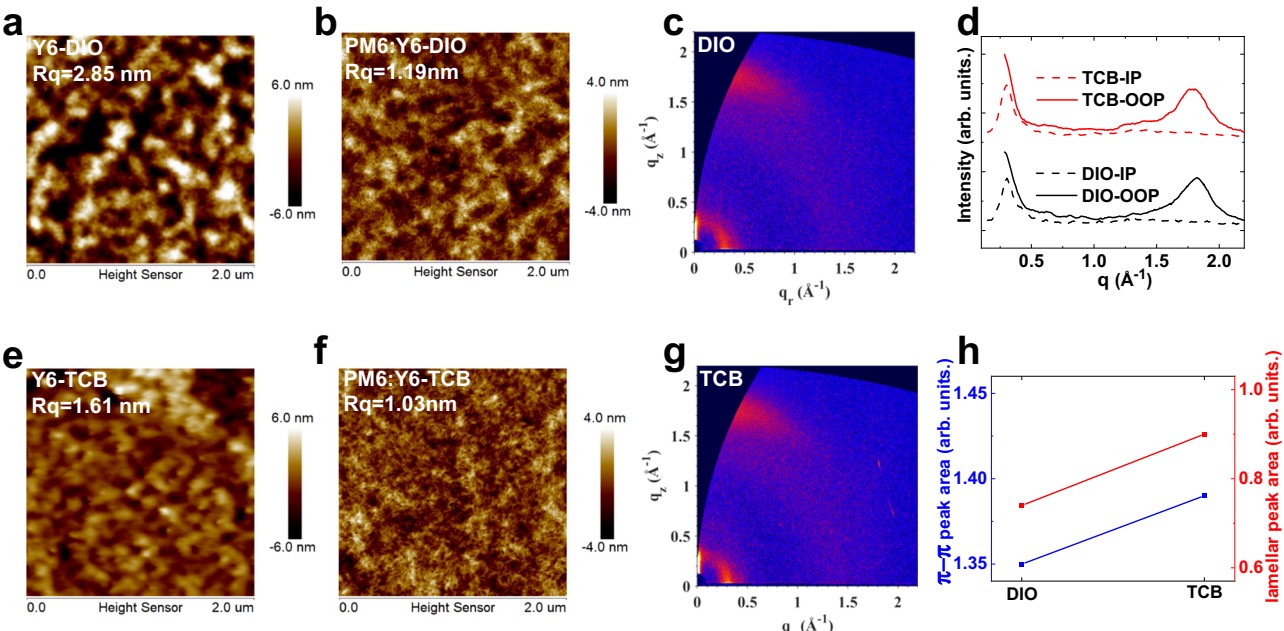

**Fig. 3 | Morphology−Surface topography and molecular stacking.** AFM height images of Y6 films (**a**, **e**) and PM6:Y6 films (**b**, **f**) with DIO and TCB treatment. 2D GIWAXS diffraction patterns (**c**, **g**) and 1D GIWAXS diffraction patterns (**d**) of PM6:Y6 blend films with DIO and TCB treatment. **h** The areas of π-π and lamellar diffraction peak for PM6:Y6 blend films with DIO and TCB treatment. Source data are provided as a Source Data file.

treatment exhibits a longer decay time ($\tau = 1.82$ µs) than that of the DIO one ($\tau = 1.39$ µs), which indicates that TCB treatment is a more effective way to suppress charge carrier recombination. The free charge carrier recombination mechanism was further studied by the dependence of $V_{OC}$ on light intensity (Supplementary Fig. 5).

The power $n$ of the TCB-treated device is 1.02, lower than that of the DIO device ($n = 1.09$), underlining the DIO device suffers from more serious trap-assisted recombination and this should ascribe to the residual DIO in active layer[34–36]. Because the vapor pressure of TCB (77 Pa at 25 °C) is much higher than that of DIO (0.03 Pa at 25 °C)[37–39], TCB shows much more excellent volatility than DIO (Supplementary Fig. 6). And TCB can be removed during spin-coating process (Supplementary Fig. 7 and 8), therefore TCB-treated device suffers from less trap-assistant recombination. The combination of higher charge mobility, more balanced charge transport, and less recombination restricts charge carrier accumulation, thereby contributing to the enhanced $J_{SC}$ and FF in the TCB-treated device[40].

As DIO is the most widely used benchmark additive in pursing modern high-efficiency OSCs so far, the obviously higher $V_{OC}$ via TCB processing is of upmost interest for OSC society. To obtain more insights, we quantitatively analyzed the $V_{OC}$ loss (also known as energy loss ($E_{loss}$)) in devices processed with different treatments. The $E_{loss}$ in solar cells can normally be divided into three parts, named $\Delta E_1$, $\Delta E_2$ and $\Delta E_3$. $\Delta E_1$ and $\Delta E_2$ are related to radiative recombination above and below the bandgap, respectively[41,42]. $\Delta E_3$ is non-radiative recombination loss, also known as $E_{loss,nr}$. The calculation procedure is presented in Methods section ('The calculation procedure of $E_{loss}$' part and Supplementary Fig. 9), and detailed $E_{loss}$ parameters are summarized in Table 2 and Figs. 2e, f. The bandgaps of the DIO processed and TCB processed OSCs are 1.391 eV and 1.396 eV, respectively, corresponding to $\Delta E_1$ values of

0.261 eV and 0.262 eV, respectively. As for $\Delta E_2$, the DIO processed and TCB processed OSCs show similar values, around 0.06 eV, illustrating the nearly same charge transfer states in these two OSCs. The first method used to calculate $\Delta E_3$ is from $J$–$V$ characteristics ($\Delta E_3^{cal} = E_g − qV_{OC} − \Delta E_1 − \Delta E_2$). Here, the $\Delta E_3$ values of the DIO processed and TCB processed OSCs are 0.233 eV and 0.214 eV, respectively, meaning the more serious $E_{loss}$ in the DIO processed OSC is from non-radiative recombination loss.

In principle, $\Delta E_3$ can also be equivalently calculated from electroluminescence quantum efficiency ($EQE_{EL}$, $\Delta E_3 = E_{loss,non−rad} = −\frac{kT}{q} \ln EQE_{EL}$) according to reciprocal principle−the stronger $EQE_{EL}$, the lower $\Delta E_3$[43]. As presented in Table 2 and Fig. 2e, the TCB based device shows $EQE_{EL}$ of $3.4 \times 10^{-4}$ (corresponds to $\Delta E_3$ of 0.206 eV), while the DIO processed device shows weaker $EQE_{EL}$ of $1.7 \times 10^{-4}$ (corresponds to $\Delta E_3$ of 0.224 eV), again verifying the DIO processed OSC suffers more serious non-radiative recombination loss.

## The nanostructure and crystalline ordering of D:A blends

As device performance is directly linked to the nanostructure and crystalline ordering of D:A blend. We investigated the DIO and TCB-treated blends by tapping mode atomic force microscopy (AFM) and grazing incidence wide-angle X-ray scattering (GIWAXS). As seen in Fig. 3a, e, DIO processed Y6 films show obvious molecular aggregation, with higher root-mean-square roughness (Rq) and more hole-like "craters". In comparison, as the volatility of TCB is much more excellent, TCB is removed during spin-coating process, so the excessive molecular aggregation did not occur in TCB processed Y6 film. Although both DIO (Fig. 3b) and TCB (Fig. 3f) processed blend films show similar Rq of around 1 nm, the DIO processed film shows excessive molecular aggregation with more hole-like "craters", which is

consistent with the fact that the DIO device suffers more serious non-radiative recombination loss. Two-dimensional (2D) GIWAXS diffraction patterns are presented in Fig. 3c, g, and the relevant one-dimensional (1D) line cuts in out-of-plane (OOP) and in-plane (IP) directions are depicted in Fig. 3d. No matter with DIO or TCB processing, the blend film shows two prominent diffraction peaks, one is at around $1.8 \text{ Å}^{-1}$ due to π-π stacking in OOP direction. Since PM6 shows much weaker diffraction at $q = 1.8 \text{ Å}^{-1}$ in OOP direction than Y6 (Supplementary Fig. 10), the face-on π-π diffractions observed in blend films is more likely to stem from the π-π stacking of Y6. Another peak is at around $0.3 \text{ Å}^{-1}$ due to lamellar stacking in IP direction, reflecting preferred face-on orientations in both blends[3,29]. Detailed peak information is summarized in Fig. 3h and Supplementary Table 4. As we can see, the TCB processed film shows larger lamellar and π-π peak areas, reflecting TCB processing contributes to higher crystallinity in active blend, which is beneficial to charge transport process[7,44]. Besides, TCB can simultaneously improve the crystallinity of both polymer donor and NFA (Supplementary Fig. 10 and Supplementary Table 5), while DIO has more impact on NFA than polymer, explaining why TCB based device shows higher hole mobility.

## Non-monotonic intermediate state transition induced by TCB during film formation

To understand the working mechanisms behind these devices, we investigated the drying and crystallization dynamics of active blends by in situ GIWAXS and in situ time-resolved UV-vis reflectance spectroscopy measurements to monitor the spin-coating process in real time. The in situ GIWAXS measurement conducted at beamline 12.3.2, the Advanced Light Source, Lawrence Berkeley National Laboratory (LBNL), supports the fastest exposure of 1 frame/s, which however did not give a diffraction signal (Supplementary Fig. 11a). Although we further set an exposure time of 3 s (Supplementary Fig. 11b–f), the organic film still cannot show obvious diffraction due to the lower

crystallinity of organic molecules, unlike high crystalline inorganic material or perovskite. Extending the exposure time to dozens of seconds may help to get clear diffraction images, but as seen from the in situ UV-vis result (as we discussed later), the transition state only lasts for about 10 s. Therefore, the in situ GIWAXS setup we have access to cannot help to understand the phenomenon. Fortunately, the in situ UV-vis characterizations with resolution about 0.4 s gave us much useful information. The color mappings of normalized UV-vis reflection spectra as a function of spin-coating time for samples with DIO and TCB are presented in Fig. 4a, b, respectively. Time-resolved reflectance intensity at 600 nm (corresponds to PM6) and 750 nm (corresponds to Y6) during spin-coating process are extracted and plotted in Fig. 4c, f, respectively. As we can see, the reflectance of the DIO sample gets saturated very quickly (within 0.73 s), corresponding to the removal of host solvent, chloroform. However, the TCB sample takes more time (over 10 s) to make the reflectance of sample steady, which illustrates the self-organization process of D:A is more complicated and lasts much longer. For a more direct comparison, the evolution of normalized absorption spectra at representative time points for DIO and TCB samples are extracted in Fig. 4d, e, respectively. Unlike the DIO sample, the TCB sample shows an interesting non-monotonic intermediate state transition. Starting from liquid film at 0 s, the 0.73 s curves show the largest redshift in both the DIO and TCB-treated samples, corresponding to the enhanced molecular stacking caused by the transition from solution state to solid state and the interactions between additives and active materials. However, while the DIO film's spectra keep the same after 0.73 s, the TCB-treated sample shows a continuous edge blueshift until becomes saturated at around 10 s after the starting of spin-coating. This observation of first redshift then blueshift during the spin-coating film formation can also be visualized in pseudo-colored Fig. 4b, implying the molecular aggregation in TCB sample experienced a two-step process of first enhancing and then relaxing[26,45–47].

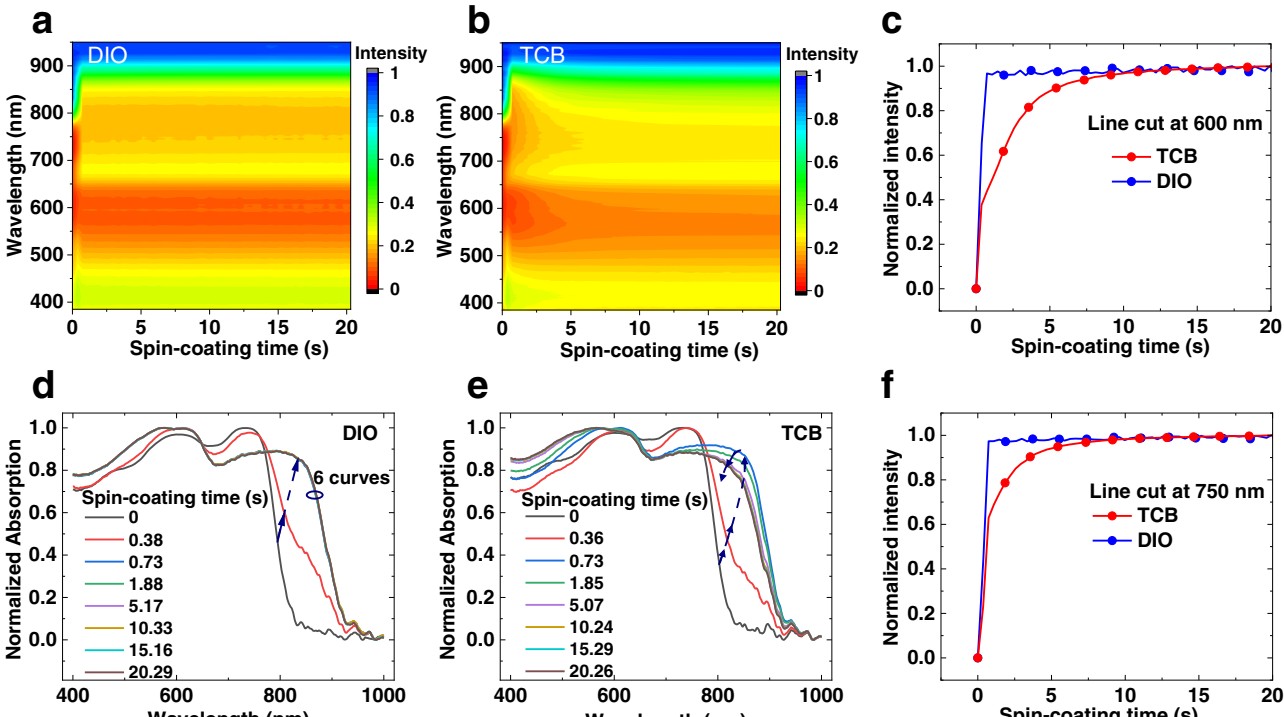

**Fig. 4 | In situ UV-vis characterization.** The color mapping of in situ UV-vis reflectance spectra as a function of spin-coating time for PM6:Y6 blends with DIO (**a**) and with TCB (**b**). Normalized in situ absorption intensity at the wavelength of 600 nm (**c**) and 750 nm (**f**) as a function of spin-coating time for PM6:Y6 blends with DIO and with TCB. Normalized absorption spectra (here we defined the absorption of sample as the difference between the reflectance of background and the reflectance of sample) at representative time points for PM6:Y6 blends with DIO (**d**) and with TCB (**e**). Source data are provided as a Source Data file.

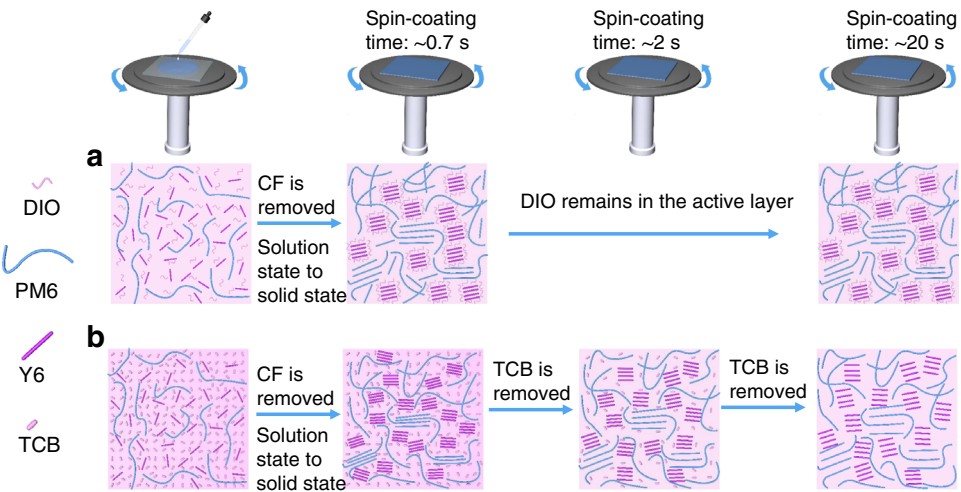

**Fig. 5 | A schematic diagram illustrating working mechanisms induced by different treatments. a** DIO treatment. **b** TCB treatment.

Based on the results from DFT simulation, TGA, in situ microscopy, FTIR, DSC GIWAXS, in situ UV-vis spectroscopy measurements and $E_{loss}$ analysis, the Fig. 5 schematic diagram illustrates the working mechanism induced by TCB. Starting from precursor solution dropping at 0 s, the solvent CF is already removed at ~0.7 s; at the same time, the wet film is converted into dry film and the interaction between additive and active materials occurs. For DIO, it can interact with NFA molecule and increase the crystallinity of NFA, thereby improving device performance[25,48]. But the low volatility of DIO tends to induce the excessive aggregation of NFA molecules, leading to increased non-radiative recombination[28]. Besides, the residual DIO is harmful to charge transport process and device stability, especially under illumination[36,48].

For TCB, it can simultaneously interact with both polymer and small molecule by hydrogen bond, thereby facilitating not only the donor polymer and small molecule NFA self-organization, but also the interpenetrating network structure. Because the volatility of TCB is excellent, it is removed during spin-coating (after the CF removal), and the interaction between TCB and active materials is released at the same time. The non-monotonic intermediate state transition then occurs, with the relaxation of molecular aggregation. This scenario explains the unique first redshift and then blueshift stages (in situ UV-vis characterization) observed in TCB case. Eventually, the TCB processing achieves film with more ordered molecular stacking, facilitating faster and more balanced charge transport. Besides, after the delicate non-monotonic intermediate state transition, the TCB-treated film exhibits more suitable molecular aggregation than the DIO-treated film, which agrees with the fact that the TCB-treated device has less non-radiative recombination than the DIO case.

**Versatility of TCB-ISM strategy**

As shown in Fig. 6, Table 3 and Supplementary Fig. 12, TCB-ISM strategy's versatility was demonstrated in five more OSC systems, including all-small-molecule system (BTR-Cl:Y6)[49], and polymer:NFA systems (PBDB-T:ITIC, PBDB-T-2Cl:IT-4F, PM1:BTP-eC9 and PM6: BTP-eC9)[22,50–52]. The same tendency as in PM6:Y6-based systems was observed: TCB processed devices show clearly improved photovoltaic performance than the benchmark DIO processed devices. Here we take the two over 19% systems as examples. The film formation processes of these two more efficient blends with DIO and TCB were investigated by in situ UV-vis characterizations, Supplementary Fig. 13 and 14 summarize the in situ UV-vis results of PM1:BTP-eC9 and PM6:BTP-eC9 blends, respectively. Like what we observed in the PM6:Y6 case, the TCB based blends show a two-step film formation process—the first enhancement and then relaxation of molecular aggregation while

the DIO based blends do not, verifying TCB can also induce the non-monotonic intermediate state transition in PM6:BTP-eC9 and PM1:BTP-eC9 blends. As a result, in PM1:BTP-eC9 systems, device efficiency increased from 17.86% by DIO processing to 19.10% by TCB processing. In PM6:BTP-eC9 systems, while the DIO device also shows an already high PCE of 17.98%, more excitingly, the TCB-ISM device offers a clearly higher PCE of 19.31%. The TCB-ISM cell was sent to an ISO/IEC 17025:2017 accredited Calibration Lab—Enli Tech Optoelectronic Calibration Lab for certification, which exhibited an efficiency of 18.93% (Supplementary Fig. 15). To the best of our knowledge, 19.31% (18.93% certified) is the highest efficiency for binary OSCs so far.

We quantitatively analyzed the $E_{loss}$ in these two more efficient OSC systems (Fig. 6c–e and Supplementary Fig. 16) and detailed $E_{loss}$ parameters were summarized in Table 4. In the PM1:BTP-eC9 system, the bandgaps of the DIO based and TCB based OSCs are 1.384 eV and 1.394 eV, respectively, corresponding to $\Delta E_1$ values of 0.259 eV and 0.262 eV, respectively. As for $\Delta E_2$, OSCs based on DIO and TCB show similar values, around 0.07 eV. As mentioned before, there are two methods to calculate $\Delta E_3$. One is from $J$–$V$ characteristics ($\Delta E_3^{cal} = E_g - qV_{OC} - \Delta E_1 - \Delta E_2$). Here, the $\Delta E_3$ of the TCB based device is 0.168 eV, the lowest reported so far in efficient (PCE ⩾ 16%) OSCs, to the best of our knowledge. Another one is from $EQE_{EL}$ ($\Delta E_3 = E_{loss,non-rad} = -\frac{kT}{q}\ln EQE_{EL}$). The DIO based OSC shows excellent high $EQE_{EL}$ of $7.2 \times 10^{-4}$, corresponding to a $\Delta E_3$ of 0.187 eV. The TCB-ISM strategy gives even more superior result, with $EQE_{EL}$ of $1.1 \times 10^{-3}$—the record in efficient OSCs (PCE ⩾ 16%) reported so far, corresponding to a $\Delta E_3$ of 0.175 eV. To the best of our knowledge, both 0.168 eV (by $J$–$V$) and 0.175 eV (by $EQE_{EL}$) are the lowest non-recombination energy loss in high efficiency OSCs (Fig. 6f and Supplementary Table 6, with reported PCE > 16%).

In the more efficient PM6:BTP-eC9 system, we observed a similar $E_{loss}$ tendency, which again verifying the effectiveness of ISM processing in suppressing non-radiative recombination loss. OSC based on PM6:BTP-eC9 with TCB-ISM processing shows a slightly higher $\Delta E_3$ of 0.19 eV, next only to that of PM1-based OSC here (Fig. 6f and Supplementary Table 6) so far. In addition to suppressing non-radiative recombination loss, ISM strategy also improves the device stability. Figure 6g shows the operational stability of PM6:BTP-eC9-based OSCs by maximum power point (MPP) tracking method. The DIO-treated device shows a stronger initial drop in PCE, suffering 17% efficiency decay within the first 75 h, while the efficiency of the TCB-treated device is only reduced by 7% within the same 75 h. After 1000-hour simulated 1-sun illumination stress test at MPP, the TCB-treated device shows very encouraging result, maintaining 78% of initial efficiency, versus 69% in the DIO case. It is worth noting the $T_{80}$ lifetime (the time

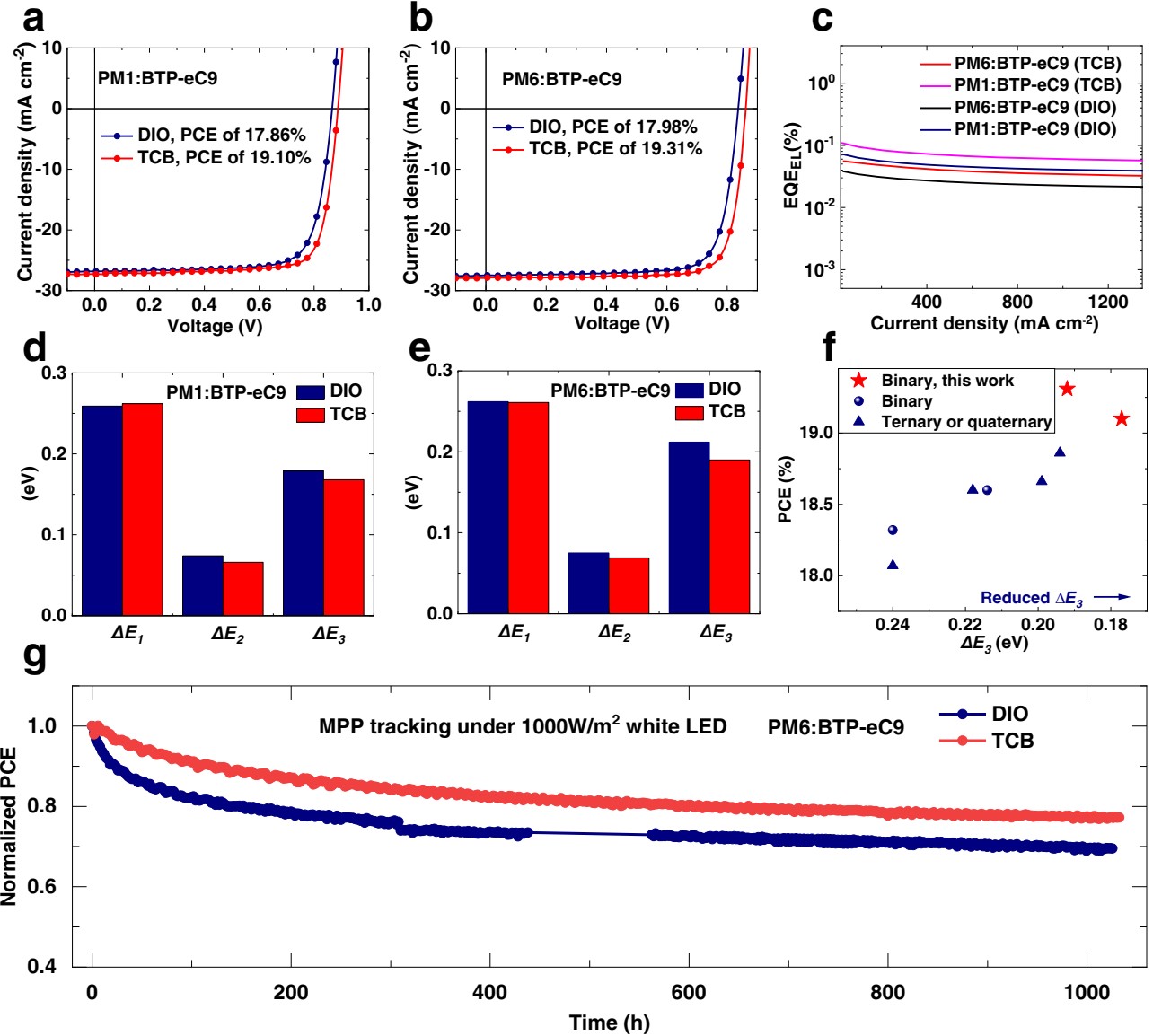

**Fig. 6 | The generality of TCB and the analysis of $V_{OC}$ loss as well as light stability. a** $J$–$V$ curves for the DIO processed and TCB processed OSCs based on PM1:BTP-eC9. **b** $J$–$V$ curves for the DIO processed and TCB processed OSCs based on PM6:BTP-eC9. **c** $EQE_{EL}$ of OSCs at various injected current densities. **d** Detailed energy loss in the DIO-processed and TCB-processed OSCs based on PM1:BTP-eC9. **e** Detailed $V_{OC}$ loss in the DIO processed and TCB processed OSCs based on PM6:BTP-eC9. **f** Comparison of PCE versus $\Delta E_3$ in reported OSCs with over 18% efficiency. **g** Light stability tests for PM6:BTP-eC9 based OSCs with different treatments, all OSCs were encapsulated and stored under continuous illumination equivalent to 1 sun in air. Source data are provided as a Source Data file.

in which device efficiency drop to 80% of initial value) of the TCB-treated device is 660 h, much higher than that of the DIO-treated one (169 h). It is also encouraging that it took 340 more hours stress test (from 660 to 1000 h) for the TCB device PCE to drop from 80% to 78% of its initial value. We believe the enhanced light stability is related to (a) TCB-induced uniform molecular aggregation for inhibiting the formation of isolated NFA aggregates as morphological traps[53,54], (b) the higher crystallinity in the TCB-treated blend for delaying the morphology evolution under light illumination[53–55], and (c) the excellent volatility of TCB for no residue left in the blend film.

## Discussion

In summary, we developed a non-monotonic intermediated state transition strategy to manipulate the BHJ OSC morphology—simultaneously optimize crystallization dynamics and energy loss of non-fullerene OSCs. Unlike the excessive molecular aggregation in films based on traditional solvent additive, the ISM strategy assists the formation of more ordered molecular stacking and suitable molecular

aggregation. As a result, we achieved obvious efficiency enhancement with reduced non-radiative recombination loss. In high-performance PM6:BTP-eC9 and PM1:BTP-eC9 binary OSC systems, the ISM strategy contributes to a record efficiency of 19.31%, and very low $E_{loss,nr}$ of 0.168 eV, respectively. The success of the ISM strategy paves a new avenue to further unleash the potential of emerging non-fullerene materials.

## Methods
### Materials

All materials are provided by commercial suppliers: PEDOT:PSS (Clevios P VP AI. 4083 (Heraeus)), PM6 (Solarmer Energy Inc.), PBDB-T (Solarmer Energy Inc.), PBDB-T-2Cl (Solarmer Energy Inc.), BTR-Cl (Solarmer Energy Inc.), PM1 (Solarmer Energy Inc.), Y6 (Solarmer Energy Inc.), ITIC (Solarmer Energy Inc.), IT-4F (Solarmer Energy Inc.), BTP-eC9 (Solarmer Energy Inc.), DIO (Tokyo Chemical Industry Co., Ltd.), TCB (Tokyo Chemical Industry Co., Ltd.), PFN-Br (Solarmer Energy Inc.), Chloroform (Sigma-Aldrich, Ltd.), methanol (Sigma-Aldrich, Ltd.) and

**Table 3 | Summary of photovoltaic operating parameters for 5 OSC systems made with different additives**

| Condition | $V_{OC}$ (V) | $J_{SC}$ (mA cm$^{-2}$) | FF (%) | PCE$^a$ (%) |
|---|---|---|---|---|
| BTR-Cl:Y6, DIO | 0.822 | 23.75 | 71.99 | 14.05 (13.82 ± 0.16) |
| BTR-Cl:Y6, TCB | 0.838 | 23.98 | 74.71 | 15.01 (14.81 ± 0.14) |
| PBDB-T:ITIC, DIO | 0.866 | 17.43 | 71.81 | 11.06 (10.81 ± 0.13) |
| PBDB-T:ITIC, TCB | 0.887 | 17.65 | 74.37 | 11.84 (11.62 ± 0.12) |
| PBDB-T-2Cl:IT-4F, DIO | 0.860 | 21.75 | 76.06 | 14.23 (13.95 ± 0.16) |
| PBDB-T-2Cl:IT-4F, TCB | 0.880 | 21.95 | 77.28 | 14.93 (14.72 ± 0.13) |
| PM1:BTP-eC9, DIO | 0.866 | 26.82 | 76.91 | 17.86 (17.51 ± 0.21) |
| PM1:BTP-eC9, TCB | 0.887 | 27.29 | 78.90 | 19.10 (18.85 ± 0.18) |
| PM6:BTP-eC9, DIO | 0.836 | 27.48 | 78.26 | 17.98 (17.65 ± 0.21) |
| PM6:BTP-eC9, TCB | 0.861 | 27.88 | 80.39 | 19.31 (19.03 ± 0.19) |
| PM6:BTP-eC9, TCB | 0.859 | 27.86 | 79.16 | 18.93$^b$ |

$^a$The average PCEs with standard deviation calculated from 20 devices. All devices were tested with a metal mask applied.
$^b$The certified photovoltaic parameters from Enli Tech Optoelectronic Calibration Lab, Accreditation Criteria: ISO/IEC 17025:2017.

**Table 4 | Detailed $E_{loss}$ parameters of two efficient OSC systems made with different additives**

| Active layer | $V_{OC}{}^c$ (V) | $E_{loss}$ (eV) | $E_g$ (eV) | $\Delta E_1$ (eV) | $\Delta E_2$ (eV) | EQE$_{EL}$ (%) | $\Delta E_3$ (eV) | $\Delta E_3{}^{cal}$ (eV) |
|---|---|---|---|---|---|---|---|---|
| PM1:BTP-eC9$^a$ | 0.872 | 0.512 | 1.384 | 0.259 | 0.074 | $7.2 \times 10^{-2}$ | 0.187 | 0.179 |
| PM1:BTP-eC9$^b$ | 0.898 | 0.496 | 1.394 | 0.262 | 0.066 | $1.1 \times 10^{-1}$ | 0.175 | 0.168 |
| PM6:BTP-ec9$^a$ | 0.845 | 0.549 | 1.394 | 0.262 | 0.075 | $3.8 \times 10^{-2}$ | 0.204 | 0.212 |
| PM6:BTP-eC9$^b$ | 0.873 | 0.521 | 1.394 | 0.262 | 0.069 | $5.7 \times 10^{-2}$ | 0.192 | 0.190 |

$^a$Devices made with DIO.
$^b$Devices made with TCB.
$^c$Device area were tested without metal mask applied.

IPA (Sigma-Aldrich, Ltd.). And all reagents and solvents are used directly without further purification.

**Device fabrication and testing**
At first, the ITO-coated glass substrates were cleaned sequentially with detergent, de-ionized water, acetone, and isopropyl alcohol (IPA) for 15 min under sonication. Then they were dried in nitrogen flow and treated with UV ozone for 30 min. The PM6:Y6, BTR-Cl:Y6, PM1:BTP-eC9, PM6:BTP-eC9 OSCs were fabricated with a conventional structure of ITO/PEDOT:PSS/active layer/PFN-Br or Phen-NaDPO/Ag. In these systems, ~50 μL PEDOT:PSS was firstly dripped on ITO substrates and spin-coated at 6000 rpm for 20 s, followed by thermal annealing on a hot plate at 120 °C for 20 min to remove the water in PEDOT:PSS film. Then, the substrates were transferred into a glovebox filled with nitrogen ($O_2 < 10$ ppm; $H_2O < 10$ ppm). The total concentrations of polymer:NFA (PN) and all-small-molecule (ASM) systems are 17 mg/mL (D:A = 1:1.2), and 16 mg/mL (D:A = 1.7:1), respectively, with chloroform as solvent. The concentrations of DIO are 0.75 % (v/v) in PM1:BTP-eC9 system and 0.5 % (v/v) in other systems. The concentration of TCB is 10 mg/mL in all OSC systems. The thickness of the active layer was controlled at around 110 nm, then the PN and ASM active layers experienced a process of thermal annealing at 100 °C for 5 min and at 110 °C for 10 min, respectively. The next stage is to coat electron transport material, ~5 nm PFN-Br, and 5 nm Phen-NaDPO were coated on the top of PN active layers and ASM active layers, respectively. Finally, these semi-finished cells were transferred into a thermal evaporation chamber with a base pressure of ~$2 \times 10^{-4}$Pa, where 100-nm Ag were deposited through a shadow mask with an active area of 11 mm$^2$. The PBDBT:ITIC and PBDB-T-2Cl:IT-4F OSCs were fabricated with an inverted structure of ITO/ZnO/active layer/MoO$_3$/Ag. In these systems, ~L ZnO precursor solution was firstly dripped on ITO substrates and spin-coated at 3000 rpm for 30 s, followed by thermal annealing on a hot plate at 200 °C for 30 min. Then, the substrates were transferred into a glovebox filled with nitrogen ($O_2 < 10$ ppm; $H_2O < 10$ ppm). The total concentrations of these two systems are 22 mg/mL (D:A = 1:1.2), with chlorobenzene as solvent.

The concentration of DIO is 1% (v/v) and the concentration of TCB is 15 mg/mL. The thickness of the active layer was controlled at around 110 nm, then the active layers experienced a process of thermal annealing at 100 °C for 5 min. Finally, these semi-finished cells were transferred into a thermal evaporation chamber with a base pressure of ~$2 \times 10^{-4}$ Pa, where 8 nm MoO$_3$ and 100 nm Ag were deposited. All devices were tested with a metal mask whose area is ~6.1 mm$^2$. The current density–voltage ($J$–$V$) curves of OSCs were tested by a Keithley 2400 source meter and an AAA grade solar simulator (SS-F7-3A, Enli Tech. Co., Ltd., Taiwan) along with AM 1.5 G spectra whose intensity was corrected by a standard silicon solar cell at 1000 W/m$^2$. The $J$–$V$ curves are measured in the forward direction from −0.2 to 1.2 V. The external quantum efficiency (EQE) was measured by a certified incident photon to electron conversion (IPCE) equipment (QE-R) from Enli Technology Co., Ltd.

**DSC and TGA**
The thermogravimetric analysis (TGA) was carried out on a Mettler Toledo TGA/DSC 1 thermogravimetric analyzer with a thermal balance under the protection of nitrogen. Differential scanning calorimetry (DSC) test was carried out on Thermal Analysis System DSC 3 (Mettler Toledo), and the data we used is from the second scan because the first-scan data may be influenced by other factors like residual solvents and the thermal history of the polymer.

**SCLC mobility measurements**
Electron-only devices with the structure of ITO/ZnO/PFN-Br/active layer/PFN-Br/Ag and hole-only devices with the structure of ITO/MoO$_3$/active layer/ MoO$_3$/Ag are used to conduct SCLC measurements. The mobilities were determined by fitting the dark-field current density-voltage curves using the Mott-Gurney relationship, which is described in the following equation,

$$J(V) = \frac{9}{8} \varepsilon_0 \varepsilon_r \mu_0 \frac{V^2}{L^3} \tag{1}$$

where J is the current density, $\varepsilon_0$ is the permittivity of free space, $\varepsilon_r$ is the relative permittivity of the material, $\mu_0$ is the zero-field mobility, V is the effective voltage and L is the thickness of the active layer. From the plot of $J^{1/2}$ versus V, the hole and electron mobilities can be deduced.

## AFM and GIWAXS

The atomic force microscopic (AFM) images were acquired using a Bruker Dimension EDGE in tapping mode. GIWAXS measurements were carried out with a Xeuss 2.0 SAXS/WAXS laboratory beamline using a Cu X-ray source (8.05 keV, 1.54 Å) and Pilatus3R 300 K detector. The incidence angle is 0.2°.

## TPV measurements

TPV is tested under the open-circuit and 1 sun intensity background light condition to explore the photovoltage decay. The subsequent voltage decay is then recorded by the digital storage oscilloscope to directly monitor charge carrier recombination. The intensity of light is 230 μW/cm² and the wavelength of light is 520 nm. The light pulse is 10 ns. The normalized curves are easier to compare the decay time and the slower decline one is the one with a longer lifetime. The photovoltage decay kinetics of all devices follow a mono-exponential decay: $\delta V = A \exp(-t/\tau)$ where $t$ is the time, and $\tau$ is the decay time. The fitted decay time would not be affected by the A value, thus the TPV curves are normalized.

## Highly sensitive EQE and EQE$_{EL}$ measurements

Highly sensitive EQE was measured using an integrated system (PECT-600, Enlitech), where the photocurrent was amplified and modulated by a lock-in instrument. EQE$_{EL}$ measurements were performed by applying external voltage/current sources through the devices (ELCT-3010, Enlitech).

## The calculation processes of $E_{loss}$

The equations used for $E_{loss}$ calculation are described as follow:

1. Radiative recombination above the bandgap ($\Delta E_1$)

$$\Delta E_1 = E_g - qV_{OC}^{SQ} \tag{2}$$

$$V_{OC}^{SQ} = \frac{kT}{q}\ln\left(\frac{J_{SC}}{J_0^{SQ}}+1\right) = \frac{kT}{q}\ln\left(\frac{q\int_0^\infty EQE_{PV}(E)\varnothing_{AM1.5}(E)dE}{q\int_{E_g}^\infty \varnothing_{BB}(E)dE}+1\right) \tag{3}$$

$$\varnothing_{BB}(E) = \frac{2\pi}{h^3c^2}E^2 e^{-\frac{E}{kT}} \tag{4}$$

2. Radiative recombination below the bandgap ($\Delta E_2$)

$$\Delta E_2 = E_{loss,rad} = qV_{OC}^{SQ} - qV_{OC}^{rad} \tag{5}$$

$$V_{OC}^{rad} = \frac{kT}{q}\ln\left(\frac{J_{SC}}{J_0^{rad}}+1\right) = \frac{kT}{q}\ln\left(\frac{q\int_0^\infty EQE_{PV}(E)\varnothing_{AM1.5}(E)dE}{q\int_{E_0}^\infty \varnothing_{BB}(E)dE}+1\right) \tag{6}$$

3. Non-radiative recombination loss ($\Delta E_3$)

$$\Delta E_3 = E_{loss,non-rad} = -\frac{kT}{q}\ln EQE_{EL} \tag{7}$$

$$\Delta E_3^{cal} = E_g - qV_{OC} - \Delta E_1 - \Delta E_2 \tag{8}$$

where E$_g$, V$_{OC}^{SQ}$, $k$, $T$, q, $\varnothing_{BB}$, and V$_{OC}^{rad}$ are energy bandgap, Shockley-Queisser (SQ) open-circuit voltage limit, the Boltzmann constant, the temperature, the elementary charge, the black body spectrum and radiative recombination open-circuit voltage limit.

## Reporting summary

Further information on research design is available in the Nature Portfolio Reporting Summary linked to this article.

## Data availability

The data that support the findings of this study are presented in Supplementary Information. And the source data underlying Figs. 2b, 3h, 4c, f, 6a, b, g, and Supplementary Figs. 3, 5, 12, as well as Tables 1, 3 and Supplementary Tables 1 and 2 are provided in Source Date file with this paper or available from the corresponding author on request. Source data are provided with this paper.

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

## Acknowledgements

G.L. thanks the Research Grants Council of Hong Kong (GRF grant 15211320, CRF C5037-18G, SRFS RGC Senior Research Fellowship Scheme (SRFS2122-5S04)), National Science Foundation of China (NSFC 51961165102), Hong Kong Polytechnic University (the Sir Sze-yuen Chung Endowed Professorship Fund (8-8480), RISE (Q-CDA5), G-SAC5), and Guangdong-Hong Kong-Macao Joint Laboratory for Photonic-Thermal-Electrical Energy Materials and Devices (GDSTC No. 2019B121205001). T.K. thanks the German Research Foundation (DFG) for funding (Fellowship No. KO6414). M.A. acknowledges support by the U.S. Department of Energy, Office of Science, Office of Basic Energy Sciences, Materials Sciences and Engineering Division under Contract No. DE-AC02-05-CH 11231 (D2S2 program KCD2S2). Work at the Molecular Foundry was supported by the Office of Science, Office of Basic Energy Sciences, of the U.S. Department of Energy under

Contract No. DE-AC02-05-CH 11231. This research used resources from the Advanced Light Source, which is a DOE Office of Science User Facility under contract no. DE-AC02-05CH11231 (beamline 12.3.2).

## Author contributions

G.L. and J.F. conceived the study. J.F. fabricated the devices and performed most of the characterizations and analysis. P.F. conducted in situ UV-vis characterization. H.L. and X.L. performed ex-situ GISAXS measurements. S.L. assisted AFM measurements. C.H., M.A., T.K., and C.M.S.-F. facilitated and helped with in situ GIWAXS measurements at the Advanced Light Source for the revision of the paper. G.L. and Y.Y. guided the study and supervised the execution. The manuscript is prepared, revised, and finalized by J.F., Y.Y., and G.L. All authors discussed the results and commented on the manuscript.

## Competing interests

The authors declare no competing interests.
