## [Peer Review File · Nature Communications]

19.31% binary organic solar cell and low non-radiative recombination enabled by non-monotonic intermediate state transitionEditorial Note: This manuscript has been previously reviewed at another journal that is not operating a transparent peer review scheme. This document only contains reviewer comments and rebuttal letters for versions considered at *Nature Communications*.

REVIEWER COMMENTS

Reviewer #2 (Remarks to the Author):

The authors tried to answer the reviewer's concerns and most of them can be properly addressed. Moreover, the authors further improved the efficiency up to 19.31% (18.93% certified), which is quite promising, especially for the binary devices. After considering the revision made and the improved efficiency by the authors, I recommend the publication of this paper in *Nature Communications* after some revisions.

1. In Fig. 1b-d, the authors studied thermal behaviors between TCB and active materials by using differential scanning calorimetry (DSC). Is the first or second scan used here? Typically, the first heating scan in DSC is mostly used for removal of residual solvents, and erase the thermal history of the polymer. The authors should confirm that.

2. It can be seen clearly in Fig. 6g that the photostability of TCB-processed device is much better than DIO-processed device. In the response, the authors ascribed the better device stability to the excellent volatility of TCB. I think the excellent volatility of TCB can induce better molecular aggregation or crystallization, which can not explain the better photostability achieved in TCB-processed devices.

3. The title is 19.31% binary organic solar cell and extremely low non-radiative recombination loss enabled by non-monotonic intermediate state transition. I think this is not reasonable since 19.31% efficiency is achieved in PM6:BTP-eC9. However, the non-monotonic intermediate state transition is observed in PM6:Y6. I strongly suggest that the authors should perform in-situ UV-vis characterizations on PM6:BTP-eC9 and PM1:BTP-eC9 blends to check non-monotonic intermediate state transition induced by TCB during film formation.

(The changes are highlighted in red color in the revised manuscript. The comments by the editor and reviewers are in black color and our responses are in blue color.

Reviewer #2 (Remarks to the Author):

The authors tried to answer the reviewer's concerns and most of them can be properly addressed. Moreover, the authors further improved the efficiency up to 19.31% (18.93% certified), which is quite promising, especially for the binary devices. After considering the revision made and the improved efficiency by the authors, I recommend the publication of this paper in Nature Communications after some revisions.

Response:

We thank the reviewer for the positive comment on our work.

1. In Fig. 1b-d, the authors studied thermal behaviors between TCB and active materials by using differential scanning calorimetry (DSC). Is the first or second scan used here? Typically, the first heating scan in DSC is mostly used for removal of residual solvents, and erase the thermal history of the polymer. The authors should confirm that.

Response:

We thank the reviewer for the valuable comment.

As the reviewer mentioned, to rule out the influence of other factors, researchers normally don't use the first-scan DSC data. Indeed, we used the second scan in this manuscript from the beginning. To make the statement more clear, in the revised manuscript, a sentence describing the details of the DSC test has been added to "DSC and TGA" part (Methods section, page 24) as below.

"Differential scanning calorimetry (DSC) test was carried out on Thermal Analysis System DSC 3 (Mettler Toledo), and the data we used is from the second scan because the first-scan data may be influenced by other factors like residual solvents and the thermal history of the polymer."

2. It can be seen clearly in Fig. 6g that the photostability of TCB-processed device is much better than DIO-processed device. In the response, the authors ascribed the better device stability to the excellent volatility of TCB. I think the excellent volatility of TCB can induce better molecular aggregation or crystallization, which can not explain the better photostability achieved in TCB-processed devices.

Response:

We thank the reviewer for the comment.

Here, based on the characterizations we performed and previous works, we would like to give a more detailed explanation of the enhanced operational stability.

First, for the molecular aggregation issue (which we assume it's for NFA), DIO has solubility selectivity - better for NFA than donor polymer. When the major solvent CF quickly evaporated, DIO with NFA evaporates slowly and it will lead to the excessive aggregation of NFA (also supported by AFM results in **Fig.3a** and **3b** in the original manuscript). On the other hand, TCB does not have the selectivity, and the excellent volatility of TCB also does not provide time for NFA to aggregate excessively (AFM results in **Fig.3e** and **3f** in the original manuscript). The more uniform distribution of NFA molecules in the TCB blend inhibits the formation of isolated NFA aggregates that served as morphological traps which are harmful to device stability (Advanced Materials 32.16 (2020): 1908305; Advanced Materials 34.26 (2022): 2110147). Second, TCB-treated film does have better crystallinity (**Fig.3c**, **3d**, **3g**, and **3h** in the original manuscript), i.e. higher quality active layer with less defect, which turns to delay the morphology evolution under light illumination (Joule 6.3 (2022): 662-675; Advanced Materials 34.26 (2022): 2110147). Third, the residual DIO in the active layer may lead to photodegradation (see a good review article: Advanced Materials 30.33 (2018): 1707114.), while TCB cannot be detected in the active layer after spin coating due to its excellent volatility (**Fig.S7**, and **S8** in the original manuscript).

Correspondingly, the sentence “We believe the enhanced light stability is related to the excellent volatility of TCB with respect to benchmark DIO.” in the original manuscript has been expanded/replaced by “We believe the enhanced light stability is related to (a) TCB-induced uniform molecular aggregation for inhibiting the formation of isolated NFA aggregates as morphological traps^{54,55}, (b) the higher crystallinity in the TCB-treated blend for delaying the morphology evolution under light illumination⁵⁴⁻⁵⁶, and (c) the excellent volatility of TCB for no residue left in the blend film.”

3. The title is 19.31% binary organic solar cell and extremely low non-radiative recombination loss enabled by non-monotonic intermediate state transition. I think this is not reasonable since 19.31% efficiency is achieved in PM6:BTP-eC9. However, the non-monotonic intermediate state transition is observed in PM6:Y6. I strongly suggest that the authors should perform in-situ UV-vis characterizations on PM6:BTP-eC9 and PM1:BTP-eC9 blends to check non-monotonic intermediate state transition induced by TCB during film formation.

Response:

We thank the reviewer for the comment.

Following your valuable suggestion, we have performed in-situ UV-vis characterization on these two blends (see **Fig. R1** and **R2** below, also **Supplementary Fig. 13** and **14** in the revised vision) and observed the same phenomenon as in the PM6:Y6 case, which

verifies TCB indeed induce the non-monotonic intermediate state transition in PM1:BTP-eC9 and PM6:BTP-eC9 blends, connecting to the high efficiency and low non-radiative recombination loss.

Correspondingly, in the main text, several sentences describing the film formation process of PM1:BTP-eC9 and PM6:BTP-eC9 blends with different treatments have been added (page 17 and 18) as below.

“The film formation processes of these two more efficient blends with DIO and TCB were investigated by in-situ UV-vis characterizations: **Supplementary Fig. 13**, and **14** summarize the in-situ UV-vis results of PM1:BTP-eC9 and PM6: BTP-eC9 blends, respectively. Same as in the PM6:Y6 case, the TCB based blends show a two-step film formation process – the first enhancement and then relaxation of molecular aggregation, while the DIO based blends do not, verifying TCB also induces the non-monotonic intermediate state transition in PM6:BTP-eC9 and PM1:BTP-eC9 blends.”

Also to fulfill the title guidance of Nature Communications, we have revised the title to: “19.31% binary organic solar cell and low non-radiative recombination-enabled by non-monotonic intermediate state transition”. By removing the words “extremely” and “loss” in the title, we think the title is appropriate now.

Fig. R1 In situ UV-vis characterization. The color mapping of in situ UV-vis reflectance spectra as a function of spin-coating time for PM1:BPT-eC9 blends with DIO (a) and with TCB (b). Normalized absorption spectra (here we defined the absorption of sample as the difference between the reflectance of background and the

reflectance of sample) at representative time points for PM1:BPT-eC9 blends with DIO (c) and with TCB (d).

Fig. R2 In situ UV-vis characterization. The color mapping of in situ UV-vis reflectance spectra as a function of spin-coating time for PM6:BPT-eC9 blends with DIO (a) and with TCB (b). Normalized absorption spectra (here we defined the absorption of sample as the difference between the reflectance of background and the reflectance of sample) at representative time points for PM6:BPT-eC9 blends with DIO (c) and with TCB (d).

REVIEWERS' COMMENTS

Reviewer #2 (Remarks to the Author):

All the concerns have been well addressed by the authors. This paper can be published as it is.

(The changes are highlighted in red color in the revised manuscript. The comments by the editor and reviewers are in black color and our responses are in blue color.

Reviewer #2 (Remarks to the Author):

All the concerns have been well addressed by the authors. This paper can be published as it is.

Response:

We thank the reviewer for the positive comment on our work.